# Neuroinflammation in Post-Ischemic Neurodegeneration of the Brain: Friend, Foe, or Both?

**DOI:** 10.3390/ijms22094405

**Published:** 2021-04-23

**Authors:** Ryszard Pluta, Sławomir Januszewski, Stanisław J. Czuczwar

**Affiliations:** 1Laboratory of Ischemic and Neurodegenerative Brain Research, Mossakowski Medical Research Institute, Polish Academy of Sciences, PL 02-106 Warsaw, Poland; sjanuszewski@imdik.pan.pl; 2Department of Pathophysiology, Medical University of Lublin, PL 20-090 Lublin, Poland; stanislawczuczwar@umlub.pl

**Keywords:** brain ischemia, stroke, neuroinflammation, microglia, astrocytes, T lymphocytes, monocytes, platelets, macrophages, leukocytes, neutrophils

## Abstract

One of the leading causes of neurological mortality, disability, and dementia worldwide is cerebral ischemia. Among the many pathological phenomena, the immune system plays an important role in the development of post-ischemic degeneration of the brain, leading to the development of neuroinflammatory changes in the brain. After cerebral ischemia, the developing neuroinflammation causes additional damage to the brain cells, but on the other hand it also plays a beneficial role in repair activities. Inflammatory mediators are sources of signals that stimulate cells in the brain and promote penetration, e.g., T lymphocytes, monocytes, platelets, macrophages, leukocytes, and neutrophils from systemic circulation to the brain ischemic area, and this phenomenon contributes to further irreversible ischemic brain damage. In this review, we focus on the issues related to the neuroinflammation that occurs in the brain tissue after ischemia, with particular emphasis on ischemic stroke and its potential treatment strategies.

## 1. Introduction

Brain ischemia and its consequences in humans are third most frequent cause of disability in 80% of survivors [1], the second most common cause of dementia, and second leading cause of death in the world [2,3,4,5,6]. Every year brain ischemia affects 17 million people worldwide, of whom 6 million die, and the other 5 million are permanently disabled [5,7,8]. The incidence of ischemic stroke in men is around 63 per 100,000 and in women around 59 per 100,000, which suggests that men are more affected by the disease than women [8]. The risk of ischemic stroke is age-related, with about 75% of all cases occurring in patients over 64 years of age, and about 25% of cases occurring in young people, suggesting that the pathology does not only affect the elderly [8]. Worldwide, the number of post-ischemic cases is currently estimated at around 33 million [5,7]. According to forecasts, the number of cases will increase to about 77 million in 2030 [5,7]. If the trend in ischemic stroke incidence continues, there will be about 12 million deaths by 2030, 70 million people will survive a stroke, and more than 200 million disability-adjusted life years will be recorded worldwide annually [5,9]. In 2010, the annual cost of treating stroke patients in Europe was around EUR 64 billion [5]. In the UK, stroke results in therapeutic and social costs of GBP 9 billion per year, with care costs accounting for about 5% of the national health system expenditure [10].

Human and animal studies have revealed that brain ischemia/ischemic stroke are risk factors for Alzheimer’s disease [11,12,13] and vice versa [14,15]. In the first year post-stroke, 4 out of 10 cases have some degree of cognitive impairment [16]. The diagnosis of dementia immediately post-stroke is difficult due to additional deficits in both global and individual cognition, e.g., attention and processing speed, language, memory, and frontal executive functions may be impaired [9]. History of ischemic stroke has been shown to be an important risk factor for the development of dementia [4,9,11,15,17,18]. It has been shown that cerebral ischemia accelerates the onset of dementia by 10 years [19]; in 10% of cases dementia will develop soon after the first stroke, and in about 41% cases it develops after a repeated stroke [9,20]. Within 25 years of post-stroke survival, the estimated development of dementia is approximately 48% [5].

The phenomena following experimental cerebral ischemia and human ischemic stroke are under constant investigation and are revealing interesting new data. Studies conducted over the last 5 years have significantly expanded our understanding of the genetic basis of brain neurodegeneration following ischemia. It is now well known that development of ischemic brain neurodegeneration is caused by a set of genetic changes that lead to neuronal loss in an amyloid and tau protein dependent manner [21,22,23,24,25], with progressive neuroinflammation [26,27] resulting in uncontrolled irreversible brain atrophy [17,28,29,30] with the development of full-blown dementia [31,32,33]. Disruption of blood supply to the brain causes neuronal death and, consequently, brain atrophy with progressive dementia. Disruption of many pathways, including oxidative stress, excitotoxicity, neuroinflammation, blood–brain barrier permeability, and others, at least partially explains post-ischemic neurodegeneration of the brain. Post-ischemic damage to neurons causes a significant release of glutamate, leading to over-activation of N-methyl-D-aspartate (NMDA) receptors and a massive Ca^2+^ inflow to neurons, resulting in their death [34]. As a result of ischemia, neurons and astrocytes produce reactive oxygen species (ROS), and the same mechanism reduces glutathione, an essential antioxidant that prevents DNA damage from ROS [35]. Irreversibly damaged brain cells and their remains, without the presence of microorganisms, trigger neuroinflammation after cerebral ischemia [8,36]. Post-ischemic oxidative stress and inflammatory processes, inter alia, cause additional damage to the blood–brain barrier and enable activated blood immune cells, such as T-lymphocytes, platelets, and neutrophils, to reach the ischemic site of the brain [8,37,38]. After the accumulation of activated immune cells from the blood in the ischemic areas of the brain, microglial cells are activated as a result of an increase in extracellular ATP after its release from the membranes of necrotic cells [39]. The activated microglia secretes pro-inflammatory factors such as cytokines and develops phagocytic properties [40]. Microglia activation has beneficial effects as it promotes the generation of growth factors such as brain-derived neurotrophic factors and removes necrotic tissue and ischemic debris, but the release of pro-inflammatory cytokines (such as tumor necrosis factor α (TNF-α)), nitric oxide, and ROS is harmful to the brain tissue after ischemia [8]. Increasing expression of cytokines promotes the expression of adhesion molecules on endothelial cells, which results in additional recruitment of, for example, leukocytes and platelets from the blood to the brain [37,38]. As neuronal death and brain tissue damage increase, there is a further increment in active microglia, infiltrating platelets, and leukocytes, resulting in more pro-inflammatory cytokines as a consequence of feedback [37]. This post-ischemic phenomenon increases both neuronal death and the infarct volume and causes poorer neurological outcomes. Neuroinflammatory changes in the brain are present in all stages of an ischemia episode, from cerebral blood flow arrest to late recirculation processes in ischemic brain tissue [26,27]. Neuroinflammation promotes further brain damage, causing the death of surviving neurons from the primary ischemia, but it also has a beneficial function to aid recovery and develop glial scaring. In this review, we look at the beneficial and harmful roles of neuroinflammation in post-ischemic brain neurodegeneration and possible future therapeutic strategies to reduce pathological responses following ischemia.

Moreover, inflammatory mechanisms are largely portrayed as deleterious to post-ischemic pathology, while in fact many immune processes such as phagocytosis help to reduce the consequences of ischemia. In this review, we strive to delineate the delicate balance between the beneficial and harmful aspects of inflammatory/immune activation in post-ischemic brain neurodegeneration, as a more detailed understanding of these processes is crucial for the development of effective therapies.

## 2. Neuroinflammation in the Post-Ischemic Brain

Numerous studies have shown an inflammatory response in brain tissue to local or complete ischemia in animals and humans [8,26,27,41,42,43,44,45]. The severity and extent of the neuroinflammation depends on the site, area, course, and type of the ischemic brain injury. Inflammation following ischemic brain injury in rats surviving 2 years after global cerebral ischemia showed different severity of microglia and astrocyte responses in different brain structures. In these animals, the study revealed significant astrocyte activation in the CA1 and CA3 areas of the hippocampus and the dentate gyrus, in the motor and sensory cortex, and in the striatum and thalamus, while microglial activation was only seen in the CA1 and CA3 areas of the hippocampus and in the motor cortex. In areas of the brain sensitive to ischemia, microglia and astrocytes showed increased activation at the same time, while in areas resistant to ischemia, only astrocytes were activated. Thus, there is strong evidence of less intense inflammation in ischemia-resistant areas of the brain. Neuroinflammatory processes are supported by microglia and astrocyte activity for up to 2 years in post-ischemic brain neurodegeneration. The study therefore revealed a chronic effect of brain ischemia on the neuroinflammatory response in the rat brain up to 2 years after the injury [27]. In another study, immunostaining confirmed the presence of T lymphocytes in the ischemic hippocampus and striatum in long-surviving animals after an ischemic episode [26]. The above observations indicate a persistent dysfunction of the blood–brain barrier, which in the long run may still allow T lymphocytes to pass from the blood to the post-ischemic brain. Such processes are supported by microglia activity up to 2 years after ischemia [27]. In addition, these animals showed increased expression of neurogenesis markers and the migration of neuroblasts in the subventricular zone [26]. Thus, the balance of degenerative processes and inflammation surveillance with neurogenesis may be decisive for long-term survival after cerebral ischemia [26]. Brain ischemia induces neuronal necrosis and apoptosis, which triggers an inflammatory response controlled by the release of ROS, cytokines, and chemokines. This process develops not only in the brain but also in the microcirculation and involves several types of cells, such as innate microglia immune cells, adaptive immune cells, and lymphocytes, enhancing neuronal death [8,41]. As a result of neuroinflammation in the brain, the secretion of many cytokines increases both in damaged brain tissue and in peripheral blood. These cytokines are involved in the progression of post-ischemic brain neurodegeneration and influence disease severity and neurological outcomes [8,41].

## 3. Pro- and Anti-Inflammatory Cytokines and Inflammatory Cells in the Post-Ischemic Brain

### 3.1. Cytokines

The occurrence of generalized inflammatory changes following brain ischemia is a relatively well-known phenomenon [46,47,48,49]. Cytokines such as IL-6 and TNF-α appear to be key mediators of this phenomenon [46,47,48,49]. Increased release of pro-inflammatory cytokines has been observed in the blood of people with ischemic stroke and has been correlated with a larger area of cerebral ischemia and worse outcomes [46,47,49]. Increased concentration of IL-6 in blood and cerebrospinal fluid is associated with an increase in neurological symptoms, a greater volume of infarction, and a worse prognosis [49]. In addition, elevated levels of TNF-α in the cerebrospinal fluid and blood in people with stroke have been associated with deteriorating neurological symptoms, an increase in infarct size, and worse clinical outcomes [48]. In contrast to the pro-inflammatory cytokines, IL-10 and IL-4 are anti-inflammatory cytokines. The exact relationship between pro- and anti-inflammatory cytokines and their relevance to clinical outcomes in ischemic stroke patients remain unexplained. However, this balance is disturbed in the early stages of an ischemic stroke [8]. This supports the study of elevated pro-inflammatory IL-6 in the blood at 12 h after cerebral ischemia in stroke patients compared to controls, and this increase has been correlated with severe neurological deficits and worse outcomes [50]. In conclusion, increased IL-6 and reduced IL-10 concentrations are present in the early stroke period and are associated with a degree of neurological deficit and stroke outcome [50]. This observation confirms the interaction between pro- and anti-inflammatory cytokines in the first phases of ischemic stroke, and the advantage of the balance in favor of inflammation results in more severe neurological deficits. Interestingly, at the moment, we cannot precisely define the phenomena modulating this interaction.

### 3.2. Cells

The involvement of cells other than neurons in the development of post-ischemic brain neurodegeneration has been evaluated in many experimental and clinical studies [1,26,27,37]. When neuroinflammation following brain ischemia begins, it starts with the release of pro-inflammatory factors that involve various cells. First, we observe the involvement of neuroglial cells in the brain. Next, leukocytes, monocytes, and other cells with immune functions enter the ischemic brain tissue. This mechanism may additionally exacerbate post-ischemic brain damage by increasing blood–brain barrier permeability, edema, and progressive neuronal death. The variety of cells involved in this process can have a beneficial or detrimental effect, depending on the post-ischemia period: early or delayed. Ultimately, we need to know which cells are involved in post-ischemic changes and when they are involved to establish ways to control them.

#### 3.2.1. Microglia

Microglia, the resident innate immune cells of the brain accounting for up to 20% of the neuroglial population, undergo morphological and phenotypic changes after brain ischemia [1]. Activated microglia function like macrophages during systemic inflammation and have the ability to remove foreign organisms and cellular debris. At rest, the microglia are referred to as small cells with wide protruding branches. However, after ischemia, microglial cells are activated, they change shape and function, but the exact mechanisms of this phenomenon are still unknown. They are activated after brain ischemia, as a result of which changes in their phenotypes can be observed [8,26,27,42,43,44,51]. Transient focal brain ischemia in the rat leads to microglia activation in the cerebral cortex of the ischemic hemisphere, and the severity and extent of the injury is reflected in the intensification of microglia activation [52]. The microglia around the post-ischemic parenchyma migrate towards the ischemic lesion and remain in close relationship with the neurons in a process called “capping”, that is, after neuron death, the capping helps in early recognition and rapid phagocytic removal of dead neurons [53,54]. They become active a few minutes after the onset of brain ischemia, increasing in number in the following days, reaching a peak on the tenth day after transient local brain ischemia [55]. After ischemic injury, the microglia activating the phenotype become amoeboid and have a functional macrophage character. After this transformation, microglia look like macrophages not only in appearance but also in their behavior, they can release cytokines and secrete extracellular matrix metalloproteinases (MMPs) that are able to damage the blood–brain barrier and thus increase its permeability. This process facilitates the early transfer of leukocytes from the circulation to the ischemic brain, contributing to an increased level of pro-inflammatory factors that aggravates post-ischemic injury. Once activated, the microglial cells can take on two different phenotypes: classic pro-inflammatory (M1) and alternative anti-inflammatory (M2). The M1 phenotype releases the cytokines TNF-α, IL-6, and IL-1β and substances with oxidizing properties, such as nitric oxide [56]. The M2 phenotype has beneficial effects, it causes ischemic brain healing post-ischemia and the release of anti-inflammatory factors, such as IL-4 and IL-10, and secretes many factors with neurotrophic properties capable of preventing the development of neuroinflammation [56]. A recent study showed that microglia depletion by the dual colony-stimulating factor-1 inhibitor, PLX3397, exacerbates brain infarction and neurological deficits [57]. Following a transient focal brain ischemia, microglial depletion enhances leukocyte infiltration, expression of inflammatory factors, and neuronal loss in mice [57]. This pathological phenomenon is dependent not only on lymphocytes and monocytes, but also on astrocyte-mediated inflammatory factors. Hence, the presence of microglial cells prevents astrocytes from secreting inflammatory factors during and after ischemia [57]. Moreover, by supporting the above, the microglial cells produce different neurotrophic factors that animate neurogenesis and plasticity [58]. Thus, following brain ischemia, different subsets of microglial cells have different roles.

#### 3.2.2. Astrocytes

Similar to microglia, astrocytes are housekeeping cells essential to the continuous functioning of the central nervous system. Astrocytes are involved in the physiological and pathological functioning of the brain. They regulate the water–ion balance; secrete neurotrophic factors; and remove unnecessary neurotransmitters, transport products, and waste of cellular metabolism. Astrocytes participate in the structure and function of the blood–brain barrier [59]. Under normal conditions, astrocytes take up excess glutamate from the extracellular space and convert it into glutamine for neurons to reuse, but during brain damage following ischemia, the degree of astrocyte damage affects their glutamate-uptake capacity [1,34,59]. How ischemia affects glutamate uptake by astrocytes is not fully elucidated, but expression of the excitatory amino acid transporter 2 (EAAT2) has been suggested to be impaired post-ischemia [60,61].

Cytokines from neurons and neuroglial cells cause post-ischemic astrocyte hyperplasia. As a result of ischemia, astrocytes release vimentin, nestin, IL-1β, monocyte chemotactic protein-1, and glial fibrillary acidic protein [27,62], which contribute to the development of reactive gliosis and the formation of glial scars after ischemia [27,63]. As a result of the Na^+^/K^+^ pump dysfunction, astrocytes swell after brain ischemia [62,64,65], which causes an increase in intracranial pressure and a consequent reduction in cerebral blood flow. Activated astrocytes release matrix metalloproteinase-2 (MMP-2) capable of damaging the extracellular matrix [66] and also contribute to the presence of ephrin-A5 in the ischemic brain area, which hinders axonal sprouting [67]. After embolic focal ischemia of the brain in rats, an exaggerated astroglial response is observed in the ischemic injury core from 4 h to 1 day, it peaks on day 4, persists for 28 days, and forms a glial scar [68]. Three days after reversible total brain ischemia in hippocampus astrocytes, significant upregulation of iNOS, glial fibrillary acidic protein (GFAP), and NADPH diaphorease expression was observed [69]. Post-mortem examination of the brain tissue after ischemia of patients who died within 7 days post-stroke showed increased expression of IL-15 in astrocytes [1]. Alternatively, IL-15 knockdown in astrocytes reduced ischemic brain damage in mice after transient local ischemia [70]. Transgenic mice expressing IL-15 with the controlled GFAP promoter exhibited increased cerebral infarction and increased neurological deficits following cerebral ischemia [1]. In addition, GFAP/Vimentin double-knockout mice showed reduced cortical blood flow in the brain and greater lesions following local ischemia [71]. Astrocytes release fibroblast growth factor-2, brain-derived neurotrophic factor, and nerve growth factor, which have neuroprotective properties [71,72]. In addition to their neurotrophic support, structurally, astrocytes by their terminal feet have a strong relationship with the endothelial cells of the brain’s capillaries and the pericytes that make up the blood–brain barrier. During brain ischemia, MMP-9 breaks the connection between the terminal feet of astrocytes and endothelial cells by degrading the basal lamina [73]. Consequently, the open blood–brain barrier acts as the main gateway for invasion of the brain by peripheral inflammatory cells.

#### 3.2.3. Neutrophils

Leukocytosis has been found to be a marker of inflammation in response to ischemic stroke. Leukocytosis is associated with a high degree of disability, impairment, and increased mortality [74]. Neutrophils are the first blood-derived immune cells to invade ischemic brain tissue, followed by monocytes. After brain ischemia, neutrophils undergo conformational changes due to the presence of numerous adhesive molecules, which facilitates their migration across the vessel wall into the brain tissue. In addition to blood-derived microglia and macrophages, neutrophils are among the most important leukocytes that infiltrate the post-ischemic brain. A high number of post-ischemic neutrophils in the brain come from the peripheral circulation. Later, neutrophils are attracted to the ischemic region by chemokines and then cause secondary damage to the ischemic tissue by releasing pro-inflammatory mediators, proteases, ROS, and MMPs [75]. These toxic mediators weaken the endothelial cell membrane and the basal lamina leading to permeability of the blood–brain barrier and the development of post-ischemic brain edema. Their onset is fairly early, reaching the brain within half an hour to several hours after ischemia, peaking over the next 3 days, and gradually decreasing over 15 days [8,76]. After five hours of recirculation, neutrophils enter the ischemically-damaged area of the brain [77,78]. Following neutrophil invasion, monocytes then adhere to the vessel wall and migrate towards the ischemic area with maximum involvement within 3–7 days after ischemia [76]. It has been noted that infiltrating neutrophils remain for more than a month in the ischemic areas of the brain and their presence is masked after 3 days by overactivation of microglia/macrophages in the inflammatory area [79]. These cells activate molecules capable of contact with the endothelial cells as early as 15 minutes after brain ischemia, and within 6–8 h they surround the brain’s blood vessels and penetrate the brain [80,81,82]. Neutrophils are believed to block microcirculation in the brain either mechanically or by secreting vasoconstrictors, releasing pro-inflammatory factors, ROS, and enzymes with hydrolytic activities [83,84,85]. In addition, neutrophils produce MMP-9, which is a protease that damages the blood–brain barrier, enhancing brain edema and causing hemorrhagic transformation of acute ischemic stroke [86]. The size of ischemic infarction and level of neurological deficits positively correlate with an increase in the number and activity of neutrophils, which in turn leads to an increased risk of death [76,87]. In contrast to neutrophils, after brain ischemia, the number of lymphocytes decreases, and thus the neutrophil/lymphocyte ratio increases. This ratio is closely related to the size of the infarct and mortality [87]. Leukocytes, which include neutrophils and T lymphocytes, intensify ischemic brain damage in many ways. First, the neutrophils adhere to the endothelium, which blocks the flow of erythrocytes through the microcirculation, which leads to the no-reflow phenomenon in the brain. Second, on the endothelial surface, activated neutrophils produce proteases, MMPs, and ROS, which significantly damage blood vessels and brain tissue. The consequence of the above phenomena is vasoconstriction and platelet aggregation inside the brain vessels [37,88]. Finally, infiltrating leukocytes further aggravate neuronal damage by activating pro-inflammatory mediators in and around the penumbra and in the core of the infarct [89].

#### 3.2.4. Lymphocytes

T lymphocytes become involved in the later stages of post-ischemic neurodegeneration of the brain. Lymphocytes surround the periphery of the ischemic lesion, and their number increases after 3 days, reaches a maximum after one week, and then decreases after another week [90]. The effect of various T lymphocytes on inflammation and thrombosis, with the consequence of increased brain damage and worsening of neurological deficits, has been demonstrated in studies in mice deficient in T lymphocytes [91,92,93]. In addition, immunodepletion of CD4^+^ lymphocyte cells in mice increased neuronal loss and was associated with more severe neurological deficits 7 days after focal brain ischemia [94]. The study showed that mice without γδ T lymphocyte cells had reduced infarct volume post-ischemia and the same phenomenon was reported in mice after administration of antibodies against the receptors of these cells [95]. Another study found an increase in CD4^+^CD28 null lymphocyte cells in stroke survivors or those who died from ischemic stroke [96]. This study revealed an association between the high probability of death after a stroke or new ischemic episodes and the CD4^+^CD28 null lymphocyte cell count, and proposed that the number of these lymphocytes could serve as a warning biomarker against recurrence of an ischemic event or death. Another study found that peripheral frequency of CD4^+^ lymphocyte cells and CD4^+^CD28 null lymphocyte cells was significantly higher in patients with acute ischemic stroke than in the control group [97]. The results of this investigation indicate that in acute ischemic stroke, a higher percentage of peripheral CD4^+^CD28 null lymphocyte cells may be associated with more massive brain damage. Analysis of this study also showed that the percentage of CD^+^CD28 null lymphocyte cells may be useful for distinguishing between subtypes of stroke. In addition, genetic study has found an increased expression of activating “pro-inflammatory” killer cell immunoglobulin-like receptor (KIR) genes in people with ischemic stroke, which likely explains the massive development of inflammation in the acute phase of stroke [98]. Accumulating evidence shows that both innate and adaptive immune cells penetrate the brain after ischemia. Up to one month after an experimental ischemic stroke, T lymphocyte cell invasion into the ischemic area has been observed and has been shown to persist for years in post-stroke patients [99]. Up to one month after focal brain ischemia, a significant increase in the number of different subtypes of T lymphocytes was observed in the peri-infarct zone [99]. T lymphocytes entering the brain after ischemia had a close interaction with activated astrocytes and a progressively developed pro-inflammatory phenotype as evidenced by markers of increased lymphocyte activation, pro-inflammatory cytokines TNF-α, INF-γ, IL-10, IL-17, and perforin, with appropriate T-bet and RORc transcription factors [99]. Treg immunodepletion using a specific CD-25 antibody aggravated tissue injury and impaired neurological deficits on day 7 after local brain ischemia in mice [94].

#### 3.2.5. Macrophages

Circulating macrophages are involved in delayed development of the neuroinflammatory mechanism in an ischemic brain. As early as 2 h after ischemia, activated macrophages can be detected in the brain [100]. Between 22 and 46 h after ischemia, both blood-born and brain-resident macrophages are dispersed throughout the ischemic injury in the brain and remain detectable for up to 1 week in mice after a 30 min ischemic injury [100]. In another study, their presence in brain tissue was recorded 4 days after the onset of ischemia, peaking after 7 days and then diminishing [55]. Pathological post-stroke mechanisms aggravate cell damage due to primary cellular events that initiate a vicious pathological cycle of inflammatory mediators that further enhances neuronal death. In summary, all the inflammatory cells described above play an important role both in initiating and enhancing the pathological response following brain ischemia, but also in maintaining homeostasis of brain cells, especially neurons that have survived a primary ischemic event.

## 4. Interaction of Inflammatory Cells in the Post-Ischemic Brain

The release of IL-6 and TNF-α by area(s) of brain ischemia enhances the invasion of neutrophils into the brain, which intensifies the blood–brain barrier permeability, and an increased number of neutrophils in the blood is associated with the extent of the infarction [101]. Moreover, the role of microglia after brain ischemia depends on the state of cell polarity. The dominance of the M1 phenotype correlates with extensive post-ischemic damage, increased anaerobic glycolysis, and activation of hypoxia-inducible factor 1α. Polarization of the microglial cells to the M1 phenotype and the resulting increased generation of IL-23 favor the recruitment and stimulation of γδ T lymphocyte cells, which play an unfavorable role within acute ischemic stroke [8]. The results of studies on experimental models of brain ischemia confirm that γδ T lymphocyte cells are pathogenic for the brain due to IL-17 secretion and stimulation of inflammatory changes [102]. Moreover, the release of anti-inflammatory cytokines by microglia such as IL-10 and TGF-β promotes the recruitment of regulatory lymphocytes that perform immunomodulatory and immunosuppressive functions in the ischemic brain [26]. Evidence supports a beneficial function of CD4^+^ lymphocyte cells in experimental brain ischemia [103]. Inflammatory cell interactions are very complex and involve intercellular crosstalk by mechanisms that are not fully understood. Evidence suggests that inflammation plays a bivalent role in the development of post-ischemic brain neurodegeneration, promoting neuronal loss and healing of post-ischemic lesions. As a consequence, neuroinflammation may be a promising target for the future treatment of stroke.

## 5. Pre-Clinical Neuroinflammation Treatment in the Post-Ischemic Brain

Anti-inflammatory actions are aimed at reducing neuroinflammatory reactions by inhibiting the factors and phenomena that increase inflammation, and at the same time stimulating anti-inflammatory factors and phenomena naturally occurring in the body (Table 1). One of the many cytokines most involved in the development of post-ischemic brain neurodegeneration is IL-1 [104]. IL-1 deficient mice showed a significant reduction in infarct volume of approximately 70% after focal brain ischemia compared to that of wild-type mice [104]. It has been shown that inhibition of microglia activation by 2% isoflurane in transient focal brain ischemia in rats reduced the infarct size, attenuated apoptosis, and significantly decreased microglia activation in ischemic penumbra [105]. Within 7 days after focal brain ischemia, edaravone, a free radical scavenger that mimics glutathione peroxidase, reduces microglia activation and early accumulation of oxidative products in rats [106]. Similarly, multiple exposures to hyperbaric oxygen reduced infarct volume by decreasing microglia activation (Table 1) [107]. Inactivation of NF-ĸB in astrocytes promoted survival of the neurons after ischemic damage in mice [108]. Treg immunodepletion using a specific CD-25 antibody aggravated tissue injury and impaired neurological deficits on day 7 after local brain ischemia in mice (Table 1) [94]. A beneficial effect of administration of a recombinant human interleukin-1 receptor antagonist on the reduction of neuronal death in rats after hypoxia-ischemia and focal cerebral ischemia has also been shown (Table 1) [109,110]. Administration of anti-TNF-α antibody (polyclonal rabbit anti-mouse TNF-α neutralizing antibody) improved neurological outcomes in rats after reversible local brain ischemia [111]. An approximately 40% reduction in infarct volume was observed in IL-10 overexpressing transgenic mice after local brain ischemia with a parallel decrease in caspase 3 levels compared to that of wild-type mice [112]. The administration of insulin-like growth factor-1 reduced infarct volume and improved sensitivity and mobility in mice [113]. In microglia after ischemia, Toll-like receptors (TLR) induce the expression of genes for cytokines with pro-inflammatory properties [114,115,116]. Knockout mice for TLR4, but not for TLR3 or TLR9, showed a significant reduction in infarct volume after ischemia compared to that of wild-type mice [114]. It has been found that deficiency of T and B lymphocyte cells in animal models of ischemic stroke resulted in a smaller lesion size and reduced neuroinflammation [117]. It has been shown that inhibition of T lymphocyte migration to the brain reduces infarct volume and post-ischemic inflammation [118]. Reduction in infarct size and improvement in neurological outcomes following local brain ischemia were noted after administration of heme oxygenase-1 and in heme oxygenase-1 overexpressing transgenic mice due to anti-inflammatory and anti-apoptotic properties [119,120]. Additionally, heme oxygenase-1 deficient mice have been shown to exhibit increased infarct size following local permanent brain ischemia compared to that of wild-type controls (Table 1) [121].

## 6. Neuroinflammation: Good or Bad?

Accumulating evidence suggests that neuroinflammation plays a key role in the pathogenesis of ischemic stroke and has become an interesting target for therapeutic intervention. Numerous reports indicate that neuroinflammatory cells play multiphase roles (beneficial and harmful) in which inhibiting the same pathway at the wrong time may exaggerate pathogenesis. Thus, better characterizing the pathophysiology of ischemic stroke together with timed treatment may provide the ultimate protective strategy of benefit.

A large amount of important data regarding the connection of inflammatory cells and their mediators that are released into the ischemic neurodegenerative brain has been accumulated in recent years. However, it is unclear whether the malfunctioning of inflammatory cells initiates the pathophysiological events of post-ischemic neurodegeneration, or whether the dysfunction of the inflammatory cells is a consequence of other adverse changes that occur in the early stages of the disease. Therefore, more data are needed on the origin of the inflammatory cell dysfunction. Recent research suggests that factors released from healthy inflammatory cells may also play a protective role when neurodegenerative conditions occur [122]. In conclusion, it is very interesting, but at the same time extremely difficult to understand how inflammatory cells exert various actions in different tissues under physiological and pathological conditions. Their functional complexity clearly requires an interdisciplinary approach to develop novel therapeutic interventions that will benefit from the multi-faceted nature of inflammatory cells, including their ability to facilitate crosstalk between the systemic environment and the brain.

Processes related to post-ischemic neurodegeneration of the brain are a strong support for the contribution of neuroinflammation to the progress of ischemic neuropathology (Figure 1). However, which aspects of this contribution seem positive or negative are still under debate. In this review, we suggest that there is a delicate balance in the response of neuroinflammation to post-ischemic brain neurodegeneration that may have both beneficial and harmful effects (Figure 1) [122]. We suggest that certain aspects of neuroinflammation in the post-ischemic brain are necessary and beneficial, and may limit or prevent post-ischemic neuropathology. By exploiting the benefits of post-ischemic neuroinflammation, we can work to find a causal therapy for post-ischemic neurodegeneration. Scientists have focused too long on the negative consequences of astrogliosis and microgliosis, judging the neuroglial cells as excited. However, recent evidence has demonstrated the enormous heterogeneity of neuroglial cells and the variability of their functioning. Although our primary focus is on the involvement of the innate immune system in post-ischemic brain neurodegeneration, evidence suggests that the adaptive immune system also plays a role in this pathology, which is an area that requires further research. Although it is clear that neuroinflammation contributes to the pathogenesis of post-ischemic injury, it is too broad a word to describe the mixture of various elements of the innate immune processes that become active as the disease progresses [122]. Microgliosis involves phagocytosis of amyloid plaques, neurofibrillary tangles, dysfunctional synapses, and the release of trophic factors for cell plasticity and growth (Figure 1) [123]. In contrast, microgliosis increases the amounts of chemokines and cytokines, which become toxic and harmful to neurons (Figure 1). Astrogliosis is also beneficial because it causes the propagation of calcium currents to increase signal conduction and improve repair and protection. In contrast, excessive astrogliosis can increase the amount of various neurotoxic substances. Therefore, the neuroinflammatory and immune responses following ischemic brain injury are both good and bad. Treatment options based on these data suggest early use of antibiotics to prevent infection (in stroke patients) and inhibition of the early inflammatory/immune response, although it may be likely that after brain ischemia develops, it is too late for this treatment, and treatment should focus on enhancing the protective immune response to accelerate the repair of neurons that survived the primary ischemic episode. It is clear that neuroinflammation is strongly involved with post-ischemic neurodegeneration of the brain (Figure 1), but a more thorough understanding of the ischemic-specific immune/inflammatory response will be critical in developing causal therapy for post-ischemic brain injury.

## 7. Conclusions

Neuroinflammation in post-ischemic brain neurodegeneration is extremely complex, with multiphase pro-inflammatory responses without causal treatment. In the case of ischemic neuronal cell damage, the locally and systemically released chemokines, cytokines, and ROS play an important and mutual role of triggering neuroinflammation in the brain. In inflammation of the nervous system, the blood–brain barrier is partially impaired and allows for increased, but not completely uncontrolled, entry of immune cells into the brain. Changes in the perivascular space and chemokine environment, coupled with impairment of the glial limitans, ultimately allow immune cells to infiltrate into the parenchyma of the brain, resulting in impaired brain function and hence exacerbation of clinical disease symptoms. An in-depth understanding of the specific pathological mechanisms used by different types of immune cells to cross the blood–brain barrier would improve therapeutic targeting by inhibiting potentially brain-damaging subsets of immune cells, while leaving the brain’s positive immune surveillance largely intact. Post-ischemic brain neurodegeneration with neuroinflammation begins in earnest when monocytes and leukocytes reach the brain, activating resident cells such as astrocytes and microglia and endothelial cells, and releasing another pool of pro-inflammatory mediators. Moreover, the evidence shows that cells involved in neuroinflammation have dual helper and deleterious functions, in fact, the same pathway, inactivated at different times, can increase or decrease ischemic damage to brain tissue. In view of the above, any future therapies should take into account the timing of their application post-ischemia. Based on the latest data, many genes can influence the course, extent of the damaged area, and prognosis in post-ischemic brain neurodegeneration. Neuroinflammation is a complex phenomenon governed by many factors that play a key role not only in the pathogenesis of post-ischemic injury, but also in determining its evolution; therefore, post-ischemic neuroinflammation may be a promising target in developing new therapeutic strategies for neurodegenerative diseases.

In summary, neuroinflammatory cells may express various activities leading eventually to either beneficial of harmful outcomes (Figure 1). No doubt, neuroinflammation is a key player in the development of ischemic neuropathology, as discussed in Section 6. A good example of the ambivalent role of neuroinflammation may be the broadly expressed pleiotropic protein osteopontin, which plays a role in neurodegenerative conditions, including Alzheimer’s disease [124]. On one hand, osteopontin is associated with detrimental effects on neurons due to recruiting inflammatory cells to the lesioned area. On the other hand, the protein may promote neuronal repair/regeneration via the inflammatory response [124]. These clearly opposed activities may be partially due to different functional domains of osteopontin that are exposed following MMP or thrombin cleavages [124].

## Figures and Tables

**Figure 1 ijms-22-04405-f001:**
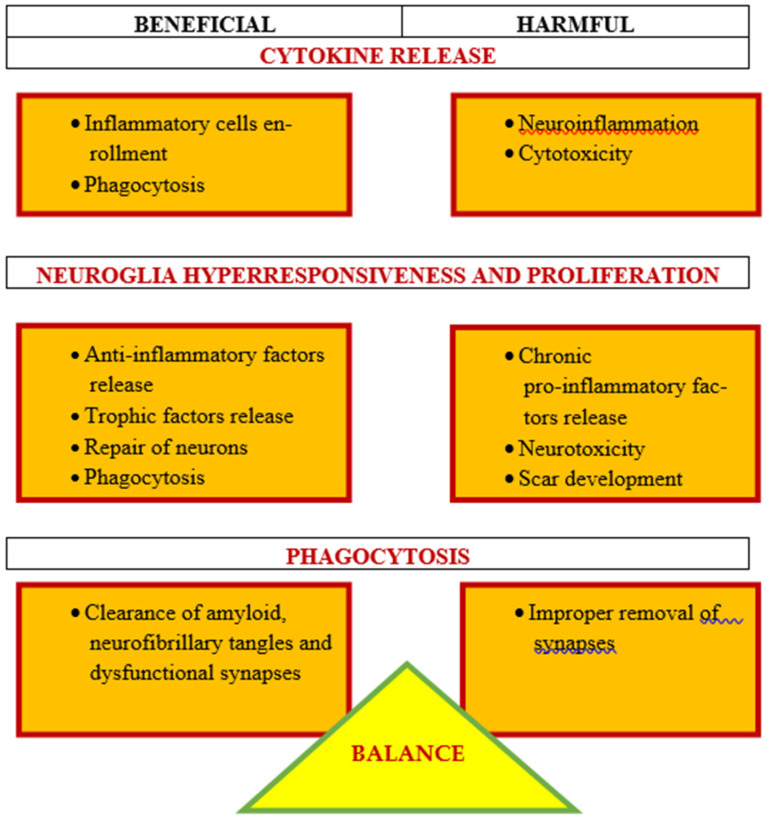
There is a delicate balance between the beneficial and harmful effects of neuroinflammation in post-ischemic brain neurodegeneration. Some neuroinflammation phenomena are protective, such as phagocytosis, but others, such as pro-inflammatory mediators, are detrimental.

**Table 1 ijms-22-04405-t001:** Preclinical studies in the prevention/treatment of neuroinflammation in post-ischemic brain neurodegeneration.

Type of Ischemia	Animal	Target	Benefits	References
Focal	Mouse	IL-1 lack	Reduction of infarct size	[104]
Focal	Rat	Microglia	Reduction of infarct size and apoptosis, microglia activation	[105]
Focal	Rat	Glutathione peroxidase	Reduction of microglia activation, oxidative products	[106]
Focal	Rat	Microglia	Reduction of infarct size	[107]
Hypoxia-ischemia	Rat	IL-1 antibody	Reduction of neurological deficits	[109]
Focal	Rat	IL-1 antibody	Reduction of neuronal death	[110]
Focal	Rat	TNF-α antibody	Improvement in neurological outcome	[111]
Permanent focal	Mouse	IL-10	Reduction of infarct size, decrease in caspase 3	[112]
Permanent focal	Mouse	IGF-1	Reduction of infarct size, improvement in sensitivity and mobility	[113]
Focal	Mouse	TLR4 lack	Reduction of infarct size	[114,115,116]
Focal	Mouse	T cells lack	Reduction of lesion size and inflammation	[117]
Permanent focal	Mouse	Lymphocytes	Reduction of lymphocytes, neuronal damage, infarct size, and inflammation	[118]
Permanent focal	Mouse	HO-1	Reduction of infarct size and neurological deficits	[119]
Focal	Mouse	HO-1	Reduction if infarct size and neurological deficits	[120]
Permanent focal	Mouse	HO-1 lack	Increase in infarct size	[121]
Focal	Mouse	NF-ĸB	Support of neuron survival	[108]
Focal	Mouse	Treg antibody	Increase of infarct size and neurological deficits	[94]
Focal	Rat	Neutrophils	Reduction of blood–brain barrier breakdown	[80]

IL-1: Interleukin 1, TNF-α: Tumor necrosis factor-α, IL-10: Interleukin 10, IGF-1: Insulin-like growth factor 1, TLR4: Toll-like receptor 4, T cells: T lymphocytes, HO-1: Heme oxygenase 1.

## Data Availability

Not applicable.

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
