# Peer review of "Neuroinflammation in Post-Ischemic Neurodegeneration of the Brain: Friend, Foe, or Both?"

_ijms, 2021, doi:10.3390/ijms22094405_

Round 1

Reviewer 1 Report

The manuscript entitled “Neuroinflammation in post-ischemic neurodegeneration of the brain: friend, foe or both?” prepared by Ryszard Pluta, SÅ‚awomir Januszewski and StanisÅ‚aw J. Czuczwar is well and clearly written. Although the paper is very interesting, there are some points to be clarified before considering the manuscript for publication. Please, see below:

  1. In my opinion, the word "therapy" is not a keyword in the context of the work as a whole.
  2. 36-38: There is a certain inconsistency in the numerical values between this sentence and the papers cited [5,7,8]. Moreover, are 17 million people - 6 million = the remaining 5 million? This sentence is unclear.
  3. 79: “NMDA” The Authors are asked to explain this abbreviation.
  4. 80: Please change “reactive oxygen species” for “reactive oxygen species (ROS)”. The authors use the abbreviation ROS later in the paper without explanation.
  5. 95: Please change “tumor necrosis factor α” for “tumor necrosis factor α (TNF-α). The authors use the abbreviation TNF-α later in the paper without explanation.
  6. 167: “In contrast to the inflammatory cytokines, IL-10 and IL-4 are anti-inflammatory cytokines.” – should be “In contrast to the pro-inflammatory cytokines…”
  7. 249: “EAAT2” The Authors are asked to explain this abbreviation.
  8. 258: Please change “matrix metalloproteinase” for “matrix metalloproteinase (MMP)”. The authors use the abbreviation MMP later in the paper without explanation.
  9. 264: “GFAP” The Authors are asked to explain this abbreviation.
  10. 387: “…by ischemic brain tissue…” - ambiguous statement, Authors are asked to correct this sentence.
  11. 435: which anti-TNF- α antibodies exactly? Please explain.
  12. 443: “TLR” The Authors are asked to explain this abbreviation.
  13. In the section "Conclusion", there is no answer to the question posed in the Title. In my opinion, it should be.
  14. Section “Pro- and anti-inflammatory cytokine in post-ischemic brain” is much less elaborate than section “Inflammatory cells in post-ischemic brain”. Is it possible to rebuild these sections and expand the section “Pro- and anti-inflammatory cytokine in post-ischemic brain”, and to compact section “Inflammatory cells in post-ischemic brain”, especially that in Table 1 both cells and inflammatory mediators are given as a target.

Author Response

The answers for the Reviewer number 1 is marked in yellow. The manuscript entitled “Neuroinflammation in post-ischemic neurodegeneration of the brain: friend, foe or both?” prepared by Ryszard Pluta, SÅ‚awomir Januszewski and StanisÅ‚aw J. Czuczwar is well and clearly written. Although the paper is very interesting, there are some points to be clarified before considering the manuscript for publication. Please, see below: 1. In my opinion, the word "therapy" is not a keyword in the context of the work as a whole. The word “therapy” was deleted from keywords. 2. 36-38: There is a certain inconsistency in the numerical values between this sentence and the papers cited [5,7,8]. Moreover, are 17 million people - 6 million = the remaining 5 million? This sentence is unclear. Certainly, this is hard to understand with “the remaining 5 million”. Now, it reads “the other 5 million”. 3. 79: “NMDA” The Authors are asked to explain this abbreviation. The abbreviation in question was explained as “N-methyl-D-aspartate”. 4. 80: Please change “reactive oxygen species” for “reactive oxygen species (ROS)”. The authors use the abbreviation ROS later in the paper without explanation. Now, “reactive oxygen species” is explained as “ROS” and this abbreviation is used throughout the whole text. 5. 95: Please change “tumor necrosis factor α” for “tumor necrosis factor α (TNF-α). The authors use the abbreviation TNF-α later in the paper without explanation. “Tumor necrosis factor α” was abbreviated as “TNF-α” throughout the text. 6. 167: “In contrast to the inflammatory cytokines, IL-10 and IL-4 are anti-inflammatory cytokines.” – should be “In contrast to the pro-inflammatory cytokines…” Done. 7. 249: “EAAT2” The Authors are asked to explain this abbreviation. Done. 8. 258: Please change “matrix metalloproteinase” for “matrix metalloproteinase (MMP)”. The authors use the abbreviation MMP later in the paper without explanation. The relevant abbreviation was added. 9. 264: “GFAP” The Authors are asked to explain this abbreviation. The abbreviation was explained (glial fibrillary acidic protein). 10. 387: “…by ischemic brain tissue…” - ambiguous statement, Authors are asked to correct this sentence. We have replaced “ischemic brain tissue” by “area(s) of brain ischemia” taken from the original paper on this issue. 11. 435: which anti-TNF- α antibodies exactly? Please explain. This was actually “polyclonal rabbit anti¬mouse TNF-a neutralizing antibody”. An explanation in the text was added. By the way, the journal’s name was corrected in ref. # 111. The correct journal is J. Cereb. Blood Flow Metab.(other details, page numbers, volume are correct). 12. 443: “TLR” The Authors are asked to explain this abbreviation. The abbreviation was explained as “toll-like receptors”. 13. In the section "Conclusion", there is no answer to the question posed in the Title. In my opinion, it should be. To be true, this issue is summarized in Section 6. However, we have added an additional para (the last para of Conclusions) on this problem, also including an additional reference [124], as requested by Reviewer #2. 14. Section “Pro- and anti-inflammatory cytokine in post-ischemic brain” is much less elaborate than section “Inflammatory cells in post-ischemic brain”. Is it possible to rebuild these sections and expand the section “Pro- and anti-inflammatory cytokine in post-ischemic brain”, and to compact section “Inflammatory cells in post-ischemic brain”, especially that in Table 1 both cells and inflammatory mediators are given as a target. Following the referee’s remark, we have come to the conclusion to put issues related to cytokines and cells into one section (Section 3). All subsequent sections we re-numbered. The authors wish to express their thanks to the reviewers for their constructive remarks. All corrections throughout the MS have been highlighted in yellow.

Reviewer 2 Report

In this review Authors analyze and discuss the role of inflammation  in post-ischemic neurodegeneration of the brain. Both protective and detrimental components of neuroinflammation are considered. The manuscript is well organized and written. Proper literature is cited and discussed with balance.

Minor point:

- Among the main players of neuroinflammation, especially considering its ambivalent role, Osteopontin (OPN) deserves to be discussed. For reference please read and cite: Cappellano G, et al. The Yin-Yang of osteopontin in nervous system diseases: damage versus repair. Neural Regen Res. 2021, 16:1131-1137. doi: 10.4103/1673-5374.300328

Author Response

The answers for the Reviewer number 2 is marked in yellow. In this review Authors analyze and discuss the role of inflammation in post-ischemic neurodegeneration of the brain. Both protective and detrimental components of neuroinflammation are considered. The manuscript is well organized and written. Proper literature is cited and discussed with balance. Minor point: Among the main players of neuroinflammation, especially considering its ambivalent role, Osteopontin (OPN) deserves to be discussed. For reference please read and cite: Cappellano G, et al. The Yin-Yang of osteopontin in nervous system diseases: damage versus repair. Neural Regen Res. 2021, 16:1131-1137. doi: 10.4103/1673-5374.300328 The requested reference was included and the role of osteopontin briefly mentioned in the last para of Conclusions. The authors wish to express their thanks to the reviewers for their constructive remarks. All corrections throughout the MS have been highlighted in yellow.